# *RNF213* c.14576G>A Is Associated with Intracranial Internal Carotid Artery Saccular Aneurysms

**DOI:** 10.3390/genes12101468

**Published:** 2021-09-23

**Authors:** Yasuo Murai, Eitaro Ishisaka, Atsushi Watanabe, Tetsuro Sekine, Kazutaka Shirokane, Fumihiro Matano, Ryuta Nakae, Tomonori Tamaki, Kenta Koketsu, Akio Morita

**Affiliations:** 1Department of Neurological Surgery, Nippon Medical School, Bunkyo-ku, Tokyo 1138603, Japan; e-ishisaka@nms.ac.jp (E.I.); kazutaka-shirokane@nms.ac.jp (K.S.); S00-078@nms.ac.jp (F.M.); kenta7240031@nms.ac.jp (K.K.); amor-tky@umin.ac.jp (A.M.); 2Division of Clinical Genetics, Kanazawa University Hospital, Kanazawa 9208640, Japan; aw3703@staff.kanazawa-u.ac.jp; 3Support Center for Genetic Medicine, Kanazawa University Hospital, Kanazawa 9208640, Japan; 4Department of Radiology, Nippon Medical School Musashi-Kosugi Hospital, Kanagawa 2118533, Japan; netti@nms.ac.jp; 5Department of Emergency and Critical Care Medicine, Nippon Medical School Hospital, Tokyo 1138603, Japan; nakae@nms.ac.jp; 6Department of Neurosurgery, Nippon Medical School Tama Nagayama Hospital, Tama, Tokyo 2068512, Japan; tamakito@nms.ac.jp

**Keywords:** cerebral aneurysm, cerebrovascular disease, genetics, stroke

## Abstract

A mutation in *RNF213* (c.14576G>A), a gene associated with moyamoya disease (>80%), plays a role in terminal internal carotid artery (ICA) stenosis (>15%) (ICS). Studies on *RNF213* and cerebral aneurysms (AN), which did not focus on the site of origin or morphology, could not elucidate the relationship between the two. However, a report suggested a relationship between *RNF213* and AN in French-Canadians. Here, we investigated the relationship between ICA saccular aneurysm (ICA-AN) and *RNF213*. We analyzed *RNF213* expression in subjects with ICA-AN and atherosclerotic ICS. Cases with a family history of moyamoya disease were excluded. AN smaller than 4 mm were confirmed as AN only by surgical or angiographic findings. *RNF213* was detected in 12.2% of patients with ICA-AN and 13.6% of patients with ICS; patients with ICA-AN and ICS had a similar risk of *RNF213* mutation expression (odds ratio, 0.884; 95% confidence interval, 0.199–3.91; *p* = 0.871). The relationship between ICA-AN and *RNF213* (c.14576G>A) was not correlated with the location of the ICA and bifurcation, presence of rupture, or multiplicity. When the etiology and location of AN were more restricted, the incidence of *RNF213* mutations in ICA-AN was higher than that reported in previous studies. Our results suggest that strict maternal vessel selection and pathological selection of AN morphology may reveal an association between genetic mutations and ICA-AN development. The results of this study may form a basis for further research on systemic vascular diseases, in which the *RNF213* (c.14576G>A) mutation has been implicated.

## 1. Introduction

The ring finger protein (*RNF)213* (c.14576G>A) mutation has been reported in 69.9–85.4% of cases of moyamoya disease (MMD) [1,2,3], an occlusive disease of the terminal portion of the internal carotid artery (ICA) [4]. Additionally, *RNF213* (c.14576G>A) has been implicated in various systemic vascular diseases, including pulmonary hypertension, peripheral pulmonary artery stenosis, and coronary stenosis [5]. In East Asian patients, *RNF213* (c.14576G>A) is also involved in 21–23.2% of cases of intracranial ICA stenosis (ICS); however, it is not observed in patients with vertebral artery stenosis [2,6]. 

The relationship between *RNF213* (c.14576G>A) and the site of intracranial vascular disease has attracted attention lately. In a study on control subjects with no intracranial vascular diseases, *RNF213* (c.14576G>A) was reported to have a high detection rate in East Asians (Japanese, 2.8%; Koreans, 2.5%; Chinese, 1.1%), but a negligible expression in Western Caucasians [2,7]. Furthermore, the frequency of *RNF213* (c.14576G>A) in Japanese patients with cerebral aneurysms (CAN) was lower than that in patients with ICS and comparable (ranging from 0 to 2.1%) to that in control subjects with no vascular lesions [1]. Notably, *RNF213* variants (p.Arg2438Cys and p.Ala2826Thr) were detected in French-Canadian patients with familial CAN [8]. 

Few studies have assessed the link between CAN and *RNF213*. Moreover, these reports failed to elucidate the details of CAN, including the site, morphology (saccular or fusiform shaped or dissected), and clinical characteristics of CAN [9]. CAN is known to exhibit different etiologies depending on the site of origin of the intracranial vessels, with different frequencies in men and women as well as different rupture rates, and thus should not be assessed in a similar way [10,11]. Hence, we aimed to investigate the role of *RNF213* in patients with non-traumatic, non-infectious, non-neoplastic ICA “saccular” aneurysms (ICAN), diagnosed as ‘intracranial ICANs’ based on surgical findings in addition to multiple imaging methods.

## 2. Materials and Methods

### 2.1. Patient Information

This study was performed in accordance with the criteria of the STROBE (Strengthening the Reporting of Observational Studies in Epidemiology) statement. This study was a case-control study conducted at hospitals affiliated with our hospital and was approved by the Ethics Committee at the institution (Approved ID; H30-26-02, 1 July 2016). Written informed consent was obtained from the study participants for blood sampling, DNA storage, and genetic analysis. As the frequency of the *RNF213* (c.14576G>A) mutation in Japanese patients with MMD has been established [1,2,6], we performed genetic analysis of the *RNF213* (c.14576G>A) mutation in cases of ICAN and atherosclerotic intracranial ICA or middle cerebral artery stenosis or occlusion (ICS). Genetic analysis of the *RNF213* (c.14576G>A) mutation was performed in patients who visited hospitals between August 2014 and August 2018 and agreed to participate in the study. A complete medical history was obtained from all patients.

### 2.2. Diagnosis

The ICAN cases included only non-traumatic, non-neoplastic, and non-infectious cases determined based on pre-existing disease and physiochemical and radiological findings. The diagnosis of ICAN was made by at least three physicians (two neurosurgeons and one radiologist), and both magnetic resonance angiography (MRA) and three-dimensional computed tomographic angiography (3DCTA) were mandatory. Infundibular dilatation [12] in the ICA was difficult to diagnose by MRA and 3DCTA alone when the size was < 4 mm. Hence, cerebral angiography or surgical findings by craniotomy were mandatory for the diagnosis of ICAN in this study. For the included cases, the location and multiplicity of ICAN, presence of rupture, and family history of CAN were noted along with other details, as previously mentioned. Patients with a family or personal history of diseases implicated in the development of CAN (including polycystic kidney disease, Ehlers–Danlos syndrome, and Marfan syndrome) were also excluded based on clinical findings. For ICS, at least two imaging studies from MRA/CTA/cerebral angiography were mandatory. Only patients with severe stenosis (>80%) from the petrous to the middle cerebral artery were included. Patients with MMD, quasi-MMD, congenital ICA deficiency, twig-like middle cerebral artery (MCA), or MCA dysplasia were excluded. Cases with a family history of MMD were excluded. Patients with a history or family history of diseases that have been indicated to be associated with MMD, including systemic lupus erythematosus, antiphospholipid antibody syndrome, periarteritis nodosa, and Sjogren’s syndrome, were also excluded from the study [4]. However, patients with atherosclerosis and diabetes mellitus were retained, despite their association with MMD [4]. Patients with a history or family history of other diseases associated with *RNF213*, including pulmonary hypertension, peripheral pulmonary artery stenosis, or coronary artery disease, were also excluded [5]. 

### 2.3. DNA Extraction and RNF213 Genotyping

Peripheral blood (5 cc) was collected from the patients and genomic DNA extraction was performed using the GENOMIX Kit (Talent, Trieste, Italy). Screening for the genotype of *RNF213* (c.14576G>A) (exon 61) was performed by small amplicon genotyping based on high resolution melting curve analysis and verified by Sanger sequencing [13]. The polymerase chain reaction (PCR) primers for c.14576G>A were as follows: forward primer, 5′-GCAAGTTGAATACAGCTCCATCA-3′, and reverse primer, 5′- TGTGCTTGCTGAGGAAGCCT-3′. The PCR conditions included an initial denaturation step at 95 °C for 2 min, 45 cycles at 94 °C for 30 s, and an annealing step at 67 °C for 30 s. After the PCR, data were collected from 55 °C–97 °C in 96-well plates by LightScanner (Idaho Technology, Salt Lake City, UT, USA) at a ramp rate of 0.101 °C/s for high-resolution melting curve analysis.

### 2.4. Data Analyses 

We performed comparative analyses between the ICAN and ICS cohorts for age at diagnosis, gender, *RNF213* (c.14576G>A) mutation, and frequency of pre-existing disease. Continuous variables showing a non-normal distribution were compared between groups using the Mann–Whitney U test. Categorical variables, such as the proportion of patients with the *RNF213* (c.14576G>A) mutation, were compared using Fisher’s exact test. A *p*-value < 0.05 was considered to indicate statistical significance. All analyses were performed using JMP 14.0 software (SAS Institute Inc., Cary, NC, USA).

## 3. Results

We assessed 47 cases of ICAN (total 69 CAN) and 22 cases of ICS during the study period. Of the 22 patients with ICS, two patients had ICAN. Hence, these two cases were included in both groups. Accordingly, 49 patients with ICAN and 22 patients with ICS were considered in the final study. The characteristics of patients in the ICAN and ICS groups are summarized in Table 1. 

The c.14576G>A mutation in *RNF213* (c.14576G>A) was found in six patients with ICAN (12.2%) and three patients with ICS (13.6%). No significant difference was noted in the frequency of *RNF213* (c.14576G>A) in the assessed groups. Patients with ICAN exhibited a similar risk of *RNF213* (c.14576G>A) mutation expression when compared with patients with ICS (odds ratio, 0.884; 95% confidence interval, 0.199–3.91; *p* = 0.871). Significantly more males (*p* = 0.0033) and cases of hypertension (*p* = 0.0038) and diabetes (*p* = 0.0046) were observed in the ICS group than in the ICAN group. The results from the analysis of characteristics in all patients with and without the *RNF213* (c.14576G>A) are summarized in Table 2. None of the assessed variables differed significantly. As MMD is a bilateral carotid artery disease, we also assessed the frequency of bilateral lesions. No significant differences were observed in the occurrence of bilateral lesions in both groups. 

The results from the analysis of patients with ICAN and ICS, with or without the *RNF213* c.14576G>A mutation, are summarized in Table 3 and Table 4. With respect to the ICAN group, no significant differences were observed between patients with and without the *RNF213* (c.14576G>A) mutation in all respects, including age, sex, past medical history, smoking history, family history of CAN, ruptured or unruptured ICAN, bilaterality, multiplicity, and location. With respect to the ICS group, no significant differences in age, gender, preexisting disease, or frequency of bilateral stenosis were noted. *RNF213* (c.14576G>A) mutation was not detected in the two cases with both ICAN and ICS.

## 4. Discussion

Our results showed that the frequency of the *RNF213* (c.14576G>A) mutation in Japanese patients with ICAN was approximately 12.5%. The frequency of *RNF213* expression was not significantly different from the frequency of ICS in patients with non-MMD, whose family history of MMD was also investigated. In contrast, a link between the site of occurrence, gender, rupture, multiplicity, or familial nature of ICAN was not observed. 

In this study, we constructed a study cohort with an emphasis on selecting patients with a reliable clinical phenotype of ICAN, no exogenous risk factors for CAN, and no family history of MMD and related disorders. The frequency of *RNF213* expression has been reported to be significantly different in the ICA and vertebrobasilar artery systems, both in stenotic and dissecting lesions of ICA (21–28%) and VBAS (0–1%) [1,2,6,14]. However, there have been no studies on *RNF213* that focus on the site of aneurysm development [8,9]. Furthermore, imaging and etiological diagnosis of small aneurysms is difficult [15]. Additionally, various causes and morphologies (e.g., trauma, dissection, and fusiform) must be reliably excluded when investigating the relationship between aneurysm development and genetic mutations. We suggested that *RNF213* mutation should be examined by strictly classifying the location and initiation factors of the CAN. 

We also examined our results for consistency with previous reports. In a survey of CAN in Japan, including CAN < 3 mm and fusiform type aneurysms, ICAN accounted for approximately 34% of all CAN [10]. Assuming that *RNF213* was involved in 12.5% of ICAN only, it would comprise less than 4.2% of all CAN cases. Our study exhibited a frequency similar to that observed previously in Japanese subjects without intracranial vascular lesions [1,2]. The investigation of the relationship between RNF213 and CAN without focusing on the cause and branching site of CAN may be a factor in the failure to detect a relationship between CAN and *RNF213* in previous studies.

The results of this study are not limited to the pathogenesis of cerebral aneurysms but may also have an impact on the search for causes of numerous diseases of systemic blood vessels. Cerebral aneurysms have been known to have characteristics, such as rupture rate differences, gender differences, and age differences depending on location and morphology, but no study has shown that there are differences in the expression of genetic mutations depending on location. In addition, there was a negative view of the relationship between aneurysms and *RNF213* itself. We clarified the relationship between aneurysms and *RNF213* by separating the aneurysm site and morphology. These results indicate that there is vascular site specificity in the functional expression of *RNF213*, which may have a ripple effect on the study of *RNF213* in other sites, where its relationship with systemic vascular diseases is also clear. If *RNF213* is detected, it may suggest which locations of blood vessels need to be watched in a systemic search for vascular diseases.

The relationship between genetic mutations and bias towards anterior circulation has been discussed. Moyamoya disease, for which *RNF213* was discovered in 2011 [16,17] to be a disease susceptibility gene, is an occlusive disease of the bilateral ICA terminations occurring from childhood. Similarly, stenosis of pulmonary arteries [18] and coronary arteries [19] is among the pathologies of blood vessels wherein *RNF213* has been implicated [5]. Of the internal carotid arteries that begin at the carotid bifurcation, the smooth muscle from the pyramidal bone to the end of the internal carotid artery is of neural crest origin [20], but the vertebral artery is not. For this reason, moyamoya disease or congenital carotid agenesis has been proposed as one of the neural crestopathies [20]. In addition, the smooth muscle of the pulmonary and coronary arteries is also of neural crest origin. Thus, it has recently become clear that the neural crest origin and the involvement of *RNF213* are the two common features of these vessels. The SOX group has been reported [21] as a transcription factor that is specific to the neural crest, and *RNF213* may play a similar role. In other words, *RNF213* may affect the formation and function of neural crest cell blood vessels in the same way that SOX10 mutations cause the loss or dysfunction of neural crest cell-derived tissues [21]. In fact, it has been shown that the *RNF213*-encoded mysterin protein is involved in the regulation of functional vascular nerve formation as a cytoplasmic AAA+ ATPase and ubiquitinating enzyme [22].

This study had some limitations. First, the number of cases assessed was small. Thus, studies with larger sample sizes and assessing posterior circulation and analyzing various clinical phenotypes, such as aneurysm morphology, age of onset, and frequency of rupture, are needed. Second, we excluded cerebral aneurysms in all intracranial sites due to ethical reasons. As the detection rate of cerebral aneurysms and intracranial stenotic lesions increases with age, future studies should be conducted in elderly patients. 

## 5. Conclusions

When the etiology and location of AN were more restricted, the incidence of *RNF213* (c.14576G>A) mutations in ICA-AN was higher than that reported in previous studies. Our results suggest that strict maternal vessel selection and pathological selection of AN morphology may reveal an association between genetic mutations and ICA-AN development. The results of this study may form a basis for further research on systemic vascular diseases, in which the *RNF213* (c.14576G>A) mutation has been implicated. Furthermore, these mutations may serve as predictive markers for patients with ICA-AN.

## Figures and Tables

**Table 1 genes-12-01468-t001:** Patient characteristics based on the disease group.

	ICAN	ICS	*p* Value
Sample size (N)	49	22	
*RNF213* +	6	3	0.573
Male: Female	11:38	13:9	0.0033
Mean Age	62.18	62.54	0.833
Standard deviation	13.82	11.18	
Standard error	1.975	2.384	
Median	65	65	
Interquartile range	20.5	19.5	
Hypertension	28	20	0.0038
Diabetes mellitus	5	9	0.0046
Dyslipidemia	20	14	0.0636
Smoking	24	14	0.1876

ICAN, internal carotid artery saccular aneurysm; ICMS, internal carotid artery or middle cerebral artery stenosis; N, number; *RNF213*, ring finger protein *213* (c.14576G>A); *RNF213* +, *RNF213* (c.14576G>A) carriers.

**Table 2 genes-12-01468-t002:** Characteristics of cases with and without *RNF213* c.14576G>A.

ICAN and ICS	*RNF* +	*RNF* −	*p* Value
N	9	62	
Male: Female	3:6	21:41	0.647
Mean Age	59.33	62.73	0.489
Standard deviation	14.87	12.77	
Standard error	4.96	1.62	
Median	65	65	
Interquartile range	29.5	19.5	
Bilateral lesions	2	12	0.755
Hypertension	7	41	0.389
Diabetes mellitus	1	13	0.431
Dyslipidemia	2	32	0.097
Smoking	4	34	0.409

ICAN, internal carotid artery saccular aneurysm; ICS, internal carotid artery or middle cerebral artery stenosis; *RNF*, ring finger protein *213* (c.14576G>A); *RNF213* +, *RNF213* (c.14576G>A) carriers; *RNF213* −, *RNF213* (c.14576G>A) non-carriers.

**Table 3 genes-12-01468-t003:** Characteristics of cases with and without *RNF213* (c.14576G>A) in internal carotid artery aneurysm.

ICAN	*RNF* +	*RNF* −	*p* Value
Number	6	43	
Mean Age	60.5	62.42	0.819
Standard deviation	17.24	13.51	
Standard error	7.04	2.06	
Median	69.5	65	
Interquartile range	33.5	20	
Male	1	10	0.592
Proximal to Paraclinoid	1	10	0.592
Proximal to SHA	1	19	0.204
Bilateral ICAN	0	6	0.436
Bilateral CAN	0	9	0.275
Multiple ICAN	2	10	0.46
Multiple CAN	2	14	0.649
Family history of CAN	1	2	0.33
Ruptured CAN	1	12	0.49
Hypertension	4	23	0.482
Diabetes mellitus	0	4	0.505
Dyslipidemia	1	18	0.204
Smoking	2	21	0.354

CAN, cerebral aneurysm; ICA, internal carotid artery; ICAN, internal carotid artery aneurysm; RNF, ring finger protein *213* (c.14576G>A); SHA, superior hypophyseal artery; *RNF213* +, *RNF213* (c.14576G>A) carriers; *RNF213* −, *RNF213* (c.14576G>A) non-carriers.

**Table 4 genes-12-01468-t004:** Characteristics of cases with and without *RNF213* (c.14576G>A) in the internal carotid artery stenosis group.

ICS	*RNF* +	*RNF* −	*p* Value
Number	3	19	
Mean Age	57	63.42	0.212
Standard deviation	11.36	11.21	
Standard error	6.56	2.57	
Median	62	66	
Interquartile range	21	21	
Male	2	11	0.642
Bilateral stenosis	2	3	0.117
Hypertension	3	17	0.74
Diabetes mellitus	1	8	0.642
Dyslipidemia	1	13	0.291
Smoking	2	12	0.764

ICS, internal carotid artery or middle cerebral artery stenosis; RNF, ring finger protein *213* (c.14576G>A); *RNF213* +, *RNF213* (c.14576G>A) carriers; *RNF213* −, *RNF213* (c.14576G>A) non-carriers.

## Data Availability

The data presented in this study are available in the article.

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
