# Peer review of "RNF213 c.14576G>A Is Associated with Intracranial Internal Carotid Artery Saccular Aneurysms"

_genes, 2021, doi:10.3390/genes12101468_

Round 1

Reviewer 1 Report

Please note that abbreviation RNF213 (c.14576G>A) has been inconsistently written in the manuscript, as RNF21(c.14576G>A) 3 and RNF213 c.14576G>A and kindly correct this.

In the text of the Results section of the manuscript, the authors state that the c.14576G>A mutation in RNF213 was found in two patients with ICS (line 138) while the information from Table 2 states 3 patients.

Could the authors expand the Discussion section of the paper by debating the clinical implications of their findings, and the use of RNF213(c.14576G>A) mutation detection in general? In addition, could the authors also discuss the reasons for anterior circulation predilection associated with this mutation?

Author Response

Thank you for the helpful comments and suggestions, which have helped us to significantly improve our manuscript. We have carefully reviewed the comments and have revised the manuscript accordingly. Our responses are given in a point-by-point manner below.

Point 1: Please note that abbreviation RNF213 (c.14576G>A) has been inconsistently written in the manuscript, as RNF21(c.14576G>A) 3 and RNF213 c.14576G>A and kindly correct this.

Response 1:

The entire manuscript has been revised for consistency.

Point 2: In the text of the Results section of the manuscript, the authors state that the c.14576G>A mutation in RNF213 was found in two patients with ICS (line 138) while the information from Table 2 states 3 patients.

Response 2:

We have corrected this. (L159)

Point 3: ICould the authors expand the Discussion section of the paper by debating the clinical implications of their findings, and the use of RNF213(c.14576G>A) mutation detection in general?

Response 3:

We have added some sentences to the Discussion, as suggested, regarding the clinical implications and applications. (L315-326)

Point 4: In addition, could the authors also discuss the reasons for anterior circulation predilection associated with this mutation?

Response 4:

We have also added content on the relationship between gene mutation and bias to anterior circulation as suggested.(L328-348)

Reviewer 2 Report

This study investigates potential relationship between RNF213 mutation and ICA aneurysm development.  It serves as a basis for larger sample size studies to investigate the impact of RNF213 mutation on the clinical behavior and progression of cerebrovascular stenotic and aneurysmal disease.

Author Response

Response to Reviewer 2 Comments

Thank you for the helpful comments and suggestions, which have helped us to significantly improve our manuscript. We have carefully reviewed the comments and have revised the manuscript accordingly.

Point 1: This study investigates potential relationship between RNF213 mutation and ICA aneurysm development.  It serves as a basis for larger sample size studies to investigate the impact of RNF213 mutation on the clinical behavior and progression of cerebrovascular stenotic and aneurysmal disease.

Response 1:

Thank you for the helpful comments and suggestions, which have helped us to significantly improve our manuscript. We have carefully reviewed the comments and have revised the manuscript accordingly. As you pointed out, we have mentioned that studies with larger sample sizes are needed. We have also added a suggestion regarding the clinical phenotypes that we would like to analyze when more samples are collected (Line 350-353).

This manuscript is a resubmission of an earlier submission. The following is a list of the peer review reports and author responses from that submission.